# Trust No Bot: Discovering Personal Disclosures in Human-LLM Conversations in the Wild

**Niloofar Mireshghallah**[*1]   **Maria Antoniak**[*2]   **Yash More**[*34]
**Yejin Choi**[12]   **Golnoosh Farnadi**[34]
[1]University of Washington    [2]Allen Institute for AI    [3]McGill University
[4]Mila-Quebec AI Institute
{niloofar,yejin}@cs.washington.edu,   mariaa@allenai.org,
{yash.more,farnadig}@mila.quebec

## Abstract

Measuring personal disclosures made in human-chatbot interactions can provide a better understanding of users' AI literacy and facilitate privacy research for large language models (LLMs). We run an extensive, fine-grained analysis on the personal disclosures made by real users to commercial GPT models, investigating the leakage of personally identifiable and sensitive information. To understand the contexts in which users disclose to chatbots, we develop a taxonomy of tasks and sensitive topics, based on qualitative and quantitative analysis of naturally occurring conversations. We discuss these potential privacy harms and observe that: (1) personally identifiable information (PII) appears in unexpected contexts such as in translation or code editing (48% and 16% of the time, respectively) and (2) PII detection alone is insufficient to capture the sensitive topics that are common in human-chatbot interactions, such as detailed sexual preferences or specific drug use habits. We believe that these high disclosure rates are of significant importance for researchers and data curators, and we call for the design of appropriate nudging mechanisms to help users moderate their interactions.

## 1 Introduction

Commercial chatbots based on large language models (LLMs) such as ChatGPT are used by millions of users to assist with both corporate tasks like writing emails and debugging code as well as personal tasks like generating erotic stories and editing visa applications. However, these models lack transparent controls and mechanisms through which users and researchers can track how these conversations are being used or shared (Liesenfeld et al., 2023), making it difficult to ground discussion about the harms that could ensue from accidental or intentional distribution of this data (Zhang et al., 2023b). The growing popularity of chatbots represents a concerning new loss in control by everyday users over how their data is shared, regulated, and passed on once they start interacting with these chatbots (Staab et al., 2023b; Li et al., 2023).

For example, LLMs are constantly updated on user information through feedback mechanisms such as RLHF (Ouyang et al., 2022) and supervised fine-tuning Gunel et al. (2020). These improvements can come at the cost of user privacy, as LLMS tend to memorize large amounts of data, making them prone to information leakage (Nasr et al., 2023). Outside of these models, users' conversations can be used by companies for any of the purposes for which other collected user data is used, e.g., to target advertisements and be sold to data brokers. These internal data collections are also at risk of hacks, data breaches or ransomware attacks Reshmi (2021).

We explore mentions of PII and sensitive topics in naturally occurring user-chatbot conversations using the WildChat dataset (Zhao et al., 2024), a collection of one million user-GPT

---

* Equal contribution

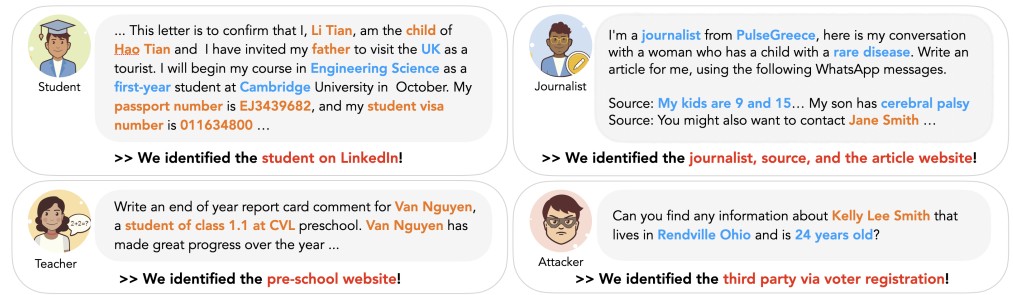

Figure 1: Real examples of disclosures we found within user-chatbot conversations in WildChat. We altered names and other PII to preserve privacy. We observe that users disclose identifiable information about themselves and others to ChatGPT and to the publicly available WildChat dataset. We were able to de-identify each of these examples.

interactions collected with user consent. Figure 1 shows a few of the many concerning sample queries that we found in this dataset. We can see that users share alarmingly sensitive information with ChatGPT (and the public WildChat dataset). To systematically analyze and draw insights from such interactions, we set out to answer the following questions:

1. What kinds of sensitive information are being shared in user-chatbot conversations?
2. What is the frequency of this leakage and how reliably can we detect it?
3. In what kinds of contexts (tasks) are different kinds and frequencies of sensitive information shared?

We build a taxonomy around the different types of sensitive information that people share, and annotate the user queries based on these categories and different PII types. While prior work has made initial progress in documenting task categories and topics in LLM-based conversations (Ouyang et al., 2023), these studies have been hampered by limited and biased access to user data, and we still know very little about the PII and other sensitive information shared in these conversations. More concretely, our main contributions include:

- An in-depth exploration of the kinds of private and sensitive information shared in user-chatbot conversations, over a series of experiments designed to illuminate when and how users reveal sensitive information.
- Automatic task and sensitive topic categorizations for 5k conversations from Wild-Chat, validated with a subset of human annotations, and novel taxonomies that capture both sensitive information and the contexts in which that information is shared. We release these annotations to support future research.[1]
- Measurements that demonstrate the limitations of PII detection systems and the frequent kinds of sensitive information that fall outside of traditional PII categories, like explicit sexual content and job applications.

Although the WildChat dataset itself has undergone one round of PII removal, we still find that over 70% of queries contain some kind of detected PII, and almost 15% mention a non-PII sensitive topic, such as sexual preferences or drug use. We also find high disclosure rates in rather surprising categories of tasks, for instance around 50% of translation queries contain some form of detected PII.

Our findings illuminate the many risks that are taken on by chatbot users. Whether these users are knowingly trading their privacy for chatbot access or are unaware that their data is being collected by chatbot companies (and the risks entailed by this collection), we believe these findings have strong implications for both chatbot designers and LLM researchers. We call for the design of appropriate nudging mechanisms to help users moderate their

---

[1] https://github.com/mireshghallah/ChatGPT-personal-disclosures

| Task | Example User Query | Detected PII | Non-Detected Sensitive Details |
|------|-------------------|--------------|-------------------------------|
| Explanation | If i want t make one glass of cannamilk. How much cannabis should i use? `i want my cannaba milk to be for microdosing` ... | *none* | drug use, personal habits |
| Generating Communications | Hello `Dan`, I just spoke with `Clement von Leigh`. `He agreed to 1.75 instead of 2.00.` Also understood that this has been communicated to `Amsterdam.` If you have any questions, please contact Clement. | first names | corporate info private email |
| Code Generation | package com.`alibaba`.adrisk.adpter.base /** * @Author: luameng * @Email: xangluameng.tangy@alibaba-inc.com * @String:`2023-05-04 15:06` */ public class OfflineQcDataDO | full name and email address | date and API access points |
| Information Retrieval | Act as an `erotic writer`. A new resident has moved into the apartment below James. Her name is Agnieska. A Polish director from multinational AI firm. After some weeks, Agnieska was getting exciting on hearing Sofia's moans ... | first names | sexual preferences |

Table 1: Examples of conversations from WildChat for a subset of our task taxonomy. We have highlighted the sensitive disclosures in yellow. See Appendix A.6 for the full set of tasks. We have altered the names and other PII in these examples.

interactions Acquisti et al. (2017), as well as increased transparency from chatbot companies. We also call for further research in local, private models and increased attention from privacy and security scholars into these high-stakes conversations.

## 2 Data and Methods

In this section, we discuss the datasets we use in the rest of this study, our sub-sampling procedure, and our annotation and taxonomy creation methods. We mainly use WildChat Zhao et al. (2024), which is a dataset of naturally occurring conversations between humans and GPT models. As a point of comparison, we also provide analysis with another dataset ShareGPT Chiang et al. (2023), which is conversations that GPT users have opted to share.

### 2.1 Data

Wildchat is a corpus of one million conversations collected by Zhao et al. (2024). The dataset includes naturally occurring human interactions with GPT-3.5 and GPT-4 models, including diverse conversations spanning many different topics. This dataset was created by providing free chatbot access to users who agreed to share their data; see §8 for ethical considerations when using this dataset. Each conversation in WildChat tracks the complete conversation thread between the user and model, and metadata including the user's hashed IP address and country are also included. We filter out the conversations that are non-English using the label provided by WildChat, as our methods rely on tools trained on English-language data. While we believe this dataset is the best resource for user-chatbot conversations openly available to researchers, this data nevertheless comes with important limitations, which we enumerate in §8. Importantly, because of the way WildChat collects its data, users might be incentivized to use WildChat for more sensitive or disallowed tasks.

### 2.2 Task Annotation

To understand the conversational contexts in which sensitive information is shared, we categorize conversations from WildChat into *tasks* representing the users' goals. We follow a bottom-up process to design a simplified set of tasks. We iteratively discuss and hand-annotate a set of 300 conversations drawn from a topic model trained on the Wildchat conversations. To train this model, we sampled the 10 conversations with the highest probability for each topic for our hand annotation, to ensure a diverse range of conversations. We trained a latent Dirichlet allocation (LDA) topic model on 10K random conversations, using the chatbot's response as the training data. We removed conversations whose prompts

had duplicate prefixes, removed punctuation, normalized numbers, and lower-cased the text. The resulting topics are shown in Appendix Table 3, along with more details about our methods.

We settled on the following 21 task categories: *summarization, model jailbreaking, prompt generation, story and script generation, song and poem generation, character description generation, code generation, code editing and debugging, communication generation, non-fictional document generation, editing text, recommendation, brainstorming, information retrieval, problem-solving, explanation, personal advice, role-playing, multiple choice questions, translation,* and *general chitchat*. We show examples in Table 1.

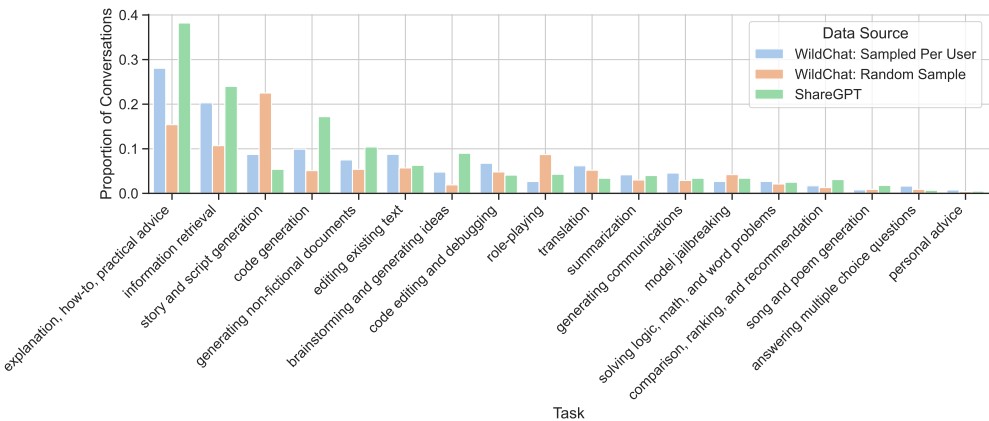

Figure 2: We plot the distribution of tasks over (a) a random sample of 5k WildChat conversations, filtered to one conversation per IP address, (b) a random sample of 1k WildChat conversations IP address or prefix filtering, and (c) a random sample of 1k ShareGPT conversations.

To avoid the costs and limitations of manually annotating a larger sample, we instead use GPT-4 (OpenAI et al., 2023) to assign task categories to a set of 5k WildChat conversations. We randomly sample conversations with the following filters: (1) we sample one conversation per hashed IP address, (2) we include only English-language conversations (as marked in the WildChat metadata), (3) we remove conversations with duplicate prefixes (the first 20 characters), and (4) we remove conversations where the user's combined turns were shorter than 20 characters. We additionally provide a comparison to (1) a similar sample of 1k WildChat conversations without the IP address and prefix filtering and (2) a random sample of 1k conversations from ShareGPT (Chiang et al., 2023). We feed each conversation to a custom zero-shot prompt template, where the conversation is formatted to show both the user and chatbot turns (see Appendix A.3) and the model is instructed to predict the task categories (more than one task can be applied to a single conversation).

To evaluate these predictions, for each task category, we sample 20 conversations predicted to include the task, and we manually verify the accuracy of the predictions, finding a mean accuracy of 89.2%. Based on this evaluation, we exclude three task categories (*general chitchat, prompt generation, generating character descriptions*) with scores below 70%.

## 2.3 Task Distribution

As shown in Figure 2, many of the WildChat queries fall in the *explanation* task, followed by *information retrieval, code generation, editing text,* and *story generation*. However, when observing the random sample without controlling for IP address, *story generation* is the most frequent task; this indicates that while *story generation* is overall the most frequent task across the conversations in WildChat, this is driven by specific power users. In contrast, we find that ShareGPT mostly contains *explanation, information retrieval,* and *code generation,* all at much higher rates than WildChat, indicating a greater skew towards these tasks in

ShareGPT that is likely caused by users selecting specific conversations to be shared in this dataset.

## 3 How much detectable PII do users share?

Our first analysis of personal disclosures is the most intuitive one: we look into the PII that the users share by running a PII detector and probing the annotations. In this section we discuss the details of this experiment and our findings.

### 3.1 PII Detection

We measure the frequency of PII in the two datasets using existing tools and taxonomies. To perform PII detection, we use the Python SDK of the commercial Azure AI language PII detection service,[2] which is designed to identify, categorize, and redact PII in unstructured text. The tool provides fine-grained annotations with over 20 different categories of PII, including organization names, URLs, banking numbers, passport numbers of different countries, etc.[3] We use this service to detect the fine-grained categories in every text in our selected subsamples of both datasets. We manually check for errors to make sure there are not high false positive rates, and we drop the erroneous categories.

### 3.2 Detected PII Distribution

Figure 3 shows the distribution of different PII entity types annotated by Azure over the WildChat and ShareGPT datasets. One noteworthy factor is that the curators of WildChat have done one round of PII removel already, using Microsoft Presidio[4]; however, Presidio is rule-based, and we find it often misses PII, especially when the PII is not well-formatted. As the histogram shows, for both datasets, most queries have some form of PII in them, with people's names and organization names taking the bulk. Overall, the distribution of PII across the two datasets seems similar, with email addresses, physical addresses, and IP addresses being the least frequent. We manually inspected these lower-count categories and observed that almost all the labels are correct, with many of them belonging to real people.

Azure AI has many categories that we dropped due to high error rates, such as national ID, passport numbers, and SWIFT code categories. However, one of the spans labeled as passport number was really a passport number. This sample is shown on the top left part of Figure 1. We have also provided more notable samples in Tables 1 and 2. Finally, Figure 4 shows a heat map of the relationship between different tasks and the detected PII, highlighting which types of information are disclosed more often, for each task. Most of the trends here are expected, with people's names being most dominant in story generation and role-playing. We also observe names in jailbreaking attempts, with numerous cases of attackers trying to extract phone numbers or personal addresses from the model. We provide an additional similar heat map in the Appendix (Figure 8), where we break down the PII categories by the country of the user.

Upon manual inspection of the IBAN category, we realized that **none** of the texts labeled as IBAN are actually international banking numbers; however we kept this category as the labeled spans were indeed PII, the majority of them being API or subscription tokens for different services, such as Telegram or analytics. Other common mistakes made by the PII detector includes labeling code and SDK calls as URLs; for example, `object.id` is labeled as a URL, which is one of the reasons that the URL count for ShareGPT is so high. Finally, another common mistake is coding constructs falling under the organization category, but the rate for this mistake is not high.

---

[2] `https://github.com/Azure/azure-sdk-for-python/tree/main/sdk/textanalytics/azure-ai-textanalytics/samples`

[3] `https://learn.microsoft.com/en-us/azure/ai-services/language-service/personally-identifiable-information/concepts/entity-categories`

[4] `https://microsoft.github.io/presidio/`

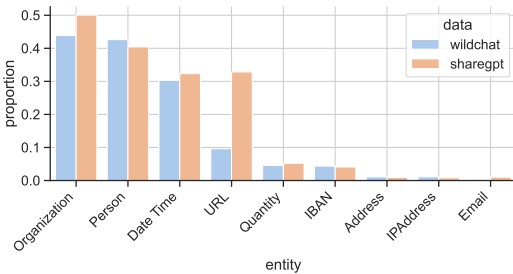

Figure 3: Fine-grained PII entities across WildChat and ShareGPT, using the Azure AI Language service for annotation. We keep the IBAN (international banking) category despite a high error rate because the detected strings are still PII (mostly API tokens).

### 3.3 Is PII detection sufficient for privacy?

While we measure frequent rates of PII in WildChat, we also observe many instances of sensitive information that is *not* captured by traditional PII detection systems. As shown in Table 1, PII detection systems are limited in the kinds of information they can detect, and many other embarrassing, identifiable (specific), and harmful information can remain undetected. For example, we observe many examples of explicit sexual content in the *story and script generation* task, which reveals private sexual preferences of the user, while the *generating communications* task often includes private text messages and emails, shared verbatim, especially related to work and finances. We also find instances of personal habits and drug use disclosed in conversations, under the *explanation and how-to category*. Motivated by these observations and prior work Brown et al. (2020); Cummings et al. (2023); Dou et al. (2023) that demonstrate disclosures can go beyond PII, we create an additional taxonomy of sensitive topics, and annotate the data accordingly, as discussed in the next section.

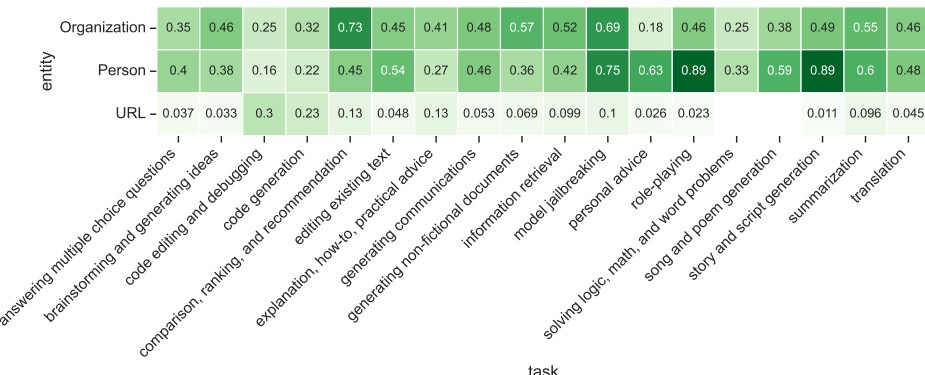

Figure 4: Relationship between task annotations of WildChat queries and detected PII.

## 4 Sensitive Topic Detection

Based on our qualitative analysis of the conversation tasks and our quantitative results in §3, we know that traditional PII categories do not capture the full range of sensitive and potentially harmful topics shared in user-chatbot conversations. In this section, we use prompting methods to extract fine-grained categories of sensitive topics, and compare measurements of those topics to PII measurements.

### 4.1 Discovering sensitive topics

We use our qualitative analysis of the conversational tasks in §2.2 as well as a review of prior work (Zhang et al., 2023b; Ouyang et al., 2023) to develop a set of categories of sensitive

topics that could potentially be harmful if revealed to the wrong audiences. These topics include academic information (e.g., asking the model to answer homework questions or generate grades for students), discussion of fandoms (i.e., discussions of television shows and book series that often reveal sexual and other preferences and hobbies and have been considered by prior work to be sensitive (Dym & Fiesler, 2018), job/visa applications, and erotic content. Table 2 shows the full list of sensitive topics with examples.

| Topic | Example User Query | % |
|---|---|---|
| Academic & Education | [recommendation letter] I am Ling Kai Associate Professor... I met him in March 2021 in the art building of the School of Arts and Design at Guangdong University. I have taught him courses such as Chinese painting basics ... He scored 76 ... | 29.9% |
| Quoted Code | line 117, in notify response = await import Optional from aiogram import types API_TOKEN = '6084658919:BAGcYQUODSWD8g0LJ8Ine6FcRZTLxg92s2q' ... ADMIN_ID_1 = 6168499378 | 19.5% |
| Fandom | Write a descriptive, fictional, imaginative screenplay of the van der linde gang reacting to an 'Elsagate' youtube video where a low quality cgi Spiderman killing a dolphin, jumping over it, then running away very slowly with a low quality walk cycle ... | 14.0% |
| Hobbies & Habits | I want for you to make an appology letter to my friend xavier beAUSE I WAS RUDETO HIM AND STOLE HIS STUFF ON MINECRAFT | 8.7% |
| Financial & Corporate | what does BLG CQBK FEE showing on HSBC bank statement mean? | 7.2% |
| Sexual & Erotic | Russian modern erotic prose, a lot of vulgar dialogue in the text, village, vegetable garden, nudity in detail, bathing naked, erotica... | 6.3% |
| Healthcare | Whats the age requirement for takind steroids in estonia? | 4.1% |
| Job, Visa, & Other Applications | Write a short and respectful mail to Indian Embassy , explaining that I Nasrin Zandi , who applied for student visa have not heard from embassy officer since Thursday when I submitted my UGC Papers , though I had called many times have not gotten a chance to speak with mr.Ronak . | 4.2% |
| Personal Relationships | my girlfriend posted a video with a boy and she tittled it #inlove with a love song and i stoped texting her am i in the wrong | 3.3% |
| Emotions & Mental Health | hi i'm feeling lonely, my parents are going through a divorce right now | 2.0% |
| Politics & Religion | how can we stop king jong un / take down north korea? | 0.7% |

Table 2: Our full taxonomy of sensitive topics along with example WildChat queries that are assigned these labels via GPT-4 annotations. We show the percent of all conversations in our 5k sample that were assigned the given task, and we highlighted sensitive information in yellow. We have altered names and other details.

As with the tasks in §2.2, we prompt GPT-4 to predict the presence of the sensitive topics; see Appendix A.4 for the prompt text. We run these predictions over the same set of 5k WildChat conversations from §2.2. We follow the same evaluation procedure as in §2.2 by hand-annotating 20 random positive predictions for each sensitive topic and discarding one sensitive topic (*quoted emails and messages*) whose accuracy fell below 70%. The mean accuracy of the rest of the topics is 87%.

## 4.2 Where does PII detection fall short?

We confirm that that PII detectors are not sufficient to detect all sensitive topics whose exposure might have harmful consequences for the user. For example, we observe in Figure 5 that PII detection systems detect many names in storytelling tasks and erotic topics, but the names in these contexts might or might not be fictional and/or sensitive. We can also see an example of this in Table 1, the first row and in Table 2. Further, Figure 7 (Appendix) shows that for many of our sensitive topics (e.g., fandom and hobbies), PII detection systems flag at best a minority of the sensitive topics. We also show the distribution of PII across different locations and countries in Figure 8 in the Appendix.

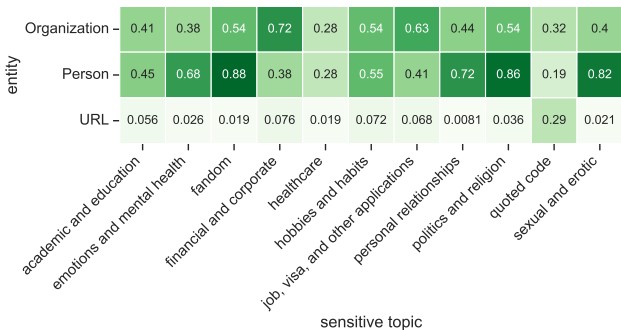

Figure 5: Relationship between sensitive topic annotations of WildChat queries and different kinds of detected PII.

### 4.3 In what conversational contexts are sensitive topics mentioned?

By comparing the task distributions with the sensitive topic distributions shown in Figure 6, we can identify the conversational contexts in which the sensitive topics are more or less likely to be mentioned, providing insights for designers of these systems. For example, we find that the model jailbreaking, role-playing, and story-generation tasks are frequent sites of *erotic* content, while role-playing, story generation, and song/poem generation are frequent sites of *fandom* mentions. The task of generating communications more often occurs with sensitive topics like *financial and corporation* information, *job and visa applications*, and *personal relationships*. These patterns can help designers develop context-specific nudges to help users protect their privacy. We also provide additional analyses of sensitive topics and tasks broken down by location of the users in Figure 8 in the Appendix.

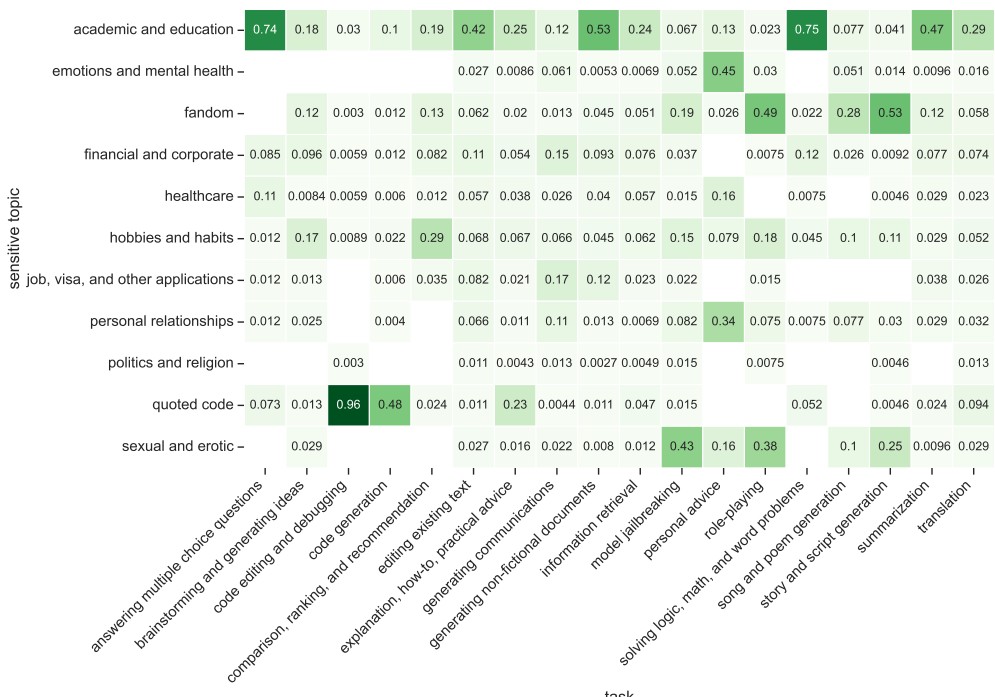

Figure 6: Relationship between sensitive topics and conversational tasks in WildChat data.

# 5  Discussion

**Design implications**  To facilitate better privacy measures there are various steps that can be enforced by the system designers, in different stages of the deployment pipeline, including data collection, training, inference and debugging (Nasr et al., 2023; Mireshghallah et al., 2022). At a minimum, data should be properly anonymized and stored safely, and chatbots based on LLMs should leverage privacy preserving methods such as differential privacy (Yu et al.; Tang et al., 2023) to limit leakage. However, better solutions that center the users' wellbeing include local models and encrypted data, and we strongly recommend such solutions over intermediate steps that prioritize user surveillance. Furthermore, users should be made aware about the data being collected as part of every interaction in the form of a nudge or disclaimer, as a part of the system design (Acquisti et al., 2017). Deployers can detect disclosures locally using light-weight methods and nudge and warn the users before the data is sent to the cloud.

Nudging can be beneficial to both users and model deployers, as it would help the users protect their data by rethinking what they share, and it can help deployers in terms of potential opt-out requests, as nudging can decrease the future retraction requests (Griesser et al., 2024; Sanchez-Rola et al., 2019). Incorporating nudges as a part of the system also helps to remind users of the sensitivity of the data being shared. To communicate the risks of sharing the data with chatbots, users should be briefed about the model training process, and how their conversations can be potentially used, e.g., for model training. System designers should provide users an easy choice to opt-in or opt-out of sharing and storing user-conversations (Gerber et al., 2023). Our work indicates that these nudges can be designed to be responsive to the user's individual task and context, perhaps by highlighting categories PII detected in the user's queries or providing a warning for certain tasks.

**Sexually explicit storytelling**  We found that an important challenge for PII detection systems for LLM prompts and outputs is dealing with storytelling. We find that a large proportion of the WildChat corpus involves story generation. Most of these queries lie either sexually explicit and/or in the fandom domain (e.g., "rewrite this TV show as if I were the main character"). These stories are full of names, ages, locations, and other text that PII detectors are likely to flag, and it would be very difficult to determine whether the user has used real names and other details in the query (especially if those details are about real people known to the user but not the user themself). And in addition to the PII, the erotic topics are themselves sensitive, as these could be embarrassing or more seriously harmful if revealed to the user's community. PII detectors will mostly not capture this sensitive information, as it is either not mentioned explicitly or falls into a category (e.g., sexual preferences) that is not usually included the training data for current PII detectors. Much prior work has either ignored or minimized the nature or frequency of these erotic stories, and we call for increased attention to this use case, as it both (a) involves serious risks to the user (both privacy risks and dependence related to increased trust and intimacy) and (b) is frequent across the dataset and often requested by the same user repeatedly.

**Relationship to self-disclosure**  The decision to self-disclose is contextual (Yang et al., 2019; Zhao et al., 2012; Li et al., 2018), and self-disclosure can be a sign of trust (Galegher et al., 1998) and growth in relationship intimacy (Altman & Taylor, 1973). When users self-disclose either about PII or about sensitive topics, this provides an indication of their level of trust with their interlocutor, and evidence suggests that users may reciprocate "disclosures" made by dialog systems (Ravichander & Black, 2018). This kind of chatbot behavior can be explicitly designed to elicit users' self-disclosures, which may be desirable for, e.g., supporting mental health or improving conversation quality (Lee et al., 2020; Ichino et al., 2022; Harmsen et al., 2023; Jo et al., 2024). Prior work has found that human-chatbot conversations can contain as much self-disclosure as human-human conversations, likely due to their perceived anonymity and lack of judgment compared to more trusted human interlocutors (Croes et al., 2024). Importantly, based on the WildChat data, it is impossible to say whether each user perceives their interlocutor in this context as the chat tool, the underlying model, the parent company, the researchers who collected WildChat, or some combination of these. More research in human-computer interaction is needed to

disentangle users' perceptions of their "relationships" with and trust in LLM-based chatbots like ChatGPT, and the design of chatbots should carefully balance features that encourage self-disclosure, application goals, and privacy concerns.

## 6    Related Work

**User-chatbot interactions**    User interactions with conversational agents (CAs) have grown in popularity over the past decade (Zheng et al., 2022; Candello et al., 2023; NAIK et al., 2023). Recent advances in LLMs have accelerated the development of CAs, making them more generalized and fluent (Ouyang et al., 2022; OpenAI et al., 2024; Park & Kulkarni, 2024). Furthermore, as LLMs perform well at a diverse of tasks(Zhao et al., 2023) like code-generation, summarization, and question-answering, they have become the go-to component for modern day chatbots and CAs.(Xu et al., 2023). In their study, Ouyang et al. (2023) analyzed ShareGPT to understand LLM-based conversational agent usage, focusing on tasks like design and planning. However, ShareGPT's lack of user consent in data collection raises authenticity issues. In contrast, our study relies on WildChat (Zhao et al., 2024), which offers a wide variety of user interactions with LLMs, and importantly, it collects data with user consent.

**Privacy risks with humans and LLMs**    Interacting with LLM-based chatbots raises significant ethical, privacy, and security concerns, necessitating careful attention to issues such as data confidentiality, user consent, and mitigation of potential biases and manipulative behaviors (Gumusel et al., 2024; Mehrotra et al., 2023) Existing work has extensively studied leakage of training data, due to memorization, in LLMs Kim et al. (2024), and how this leakage can be mitigated with different sanitization methods (Li et al., 2021; Yu et al., 2021; Cunha et al., 2021; Mireshghallah et al., 2022). Recent work has also looked at privacy risks that go beyond training data leakage (Staab et al., 2023a; Priyanshu et al., 2023; Zhang et al., 2023b; Mireshghallah et al., 2023). Our work builds on these findings by quantitatively assessing sensitive topics and PII leakage in user interactions with chatbots. Our task-based taxonomy complements the prior findings about why people talk to chat-assistants, leading to a richer understanding of disclosures.

**Self-disclosure detection**    Prior work on the detection of self-disclosures has focused on explicit disclosures statements (e.g., "My name is Maria," "I live in Seattle") (Bak et al., 2012; Ravichander & Black, 2018; Valizadeh et al., 2021; Reuel et al., 2022; Dou et al., 2023; Yang et al., 2024) rather than the implicit sensitive topics (e.g., discussion of sexually explicit topics without any personal statement) that we explore in this work. Methods for explicit self-disclosure detection have included topic modeling (Bak et al., 2014), LLM fine-tuning (Dou et al., 2023), multi-task models (Reuel et al., 2022), and LLM-based prompts (Yang et al., 2024). Other relevant work include measurements of self-disclosure in therapy conversations (Shapira & Alfi-Yogev, 2024) and conversations with dialog systems and agents (Ravichander & Black, 2018; Cho et al., 2022); the latter study revealed high rates of explicit self-disclosures, which our study (1) echoes in our detection of high rates of sensitive topics and (2) refines via task and topic categories.

## 7    Conclusion

In this work, we have studied when and how users disclose PII and sensitive topics while conversing with chatbots. We analyzed the interactions users have with LLM-based chatbots, discussed why existing PII detection methods are limited, and explained why we need better mechanisms to detect and contextualize sensitive topics. We release our novel task and sensitive topic taxonomies to the public, along with the automatic annotations using these taxonomies on our sample of the WildChat dataset. We hope that our work spurs further privacy research and brings heightened attention to the risks involved in human-chatbot conversations. *To ensure safer usage of ChatGPT and WildChat in the future, we have notified the authors of WildChat of our findings.*

# 8 Ethics Statement and Limitations

As our study illustrates, the WildChat dataset contains deeply personal self-disclosures. The sensitivity of the WildChat data has motivated our study, as we believe that researchers, practitioners, and users of LLMs all face important questions about data security. We hope that our results can help these various stakeholders develop safety guidelines, build AI literacy, and initiate further research.

WildChat was collected by using the GPT-3.5 and GPT-4 API, each of which was hosted on Hugging Face spaces and made publicly accessible (Zhao et al., 2024). The users were not required to create any account or enter any personal information to use the models. Users' consent was collected before allowing them to participate in any interactions with the model. All the users who participated in the data collection procedure were presented with a use and sharing agreement that outlines the terms for collection, usage and sharing. In exchange for signing this agreement, users received free access to models. Hashed IP addresses and country locations were publicly released with the newest version of the dataset.

The WildChat dataset provides us an opportunity to perform an in-depth study of user safety when interacting with large language models. As the conversations are real-world, our analysis captures the sensitivity of information as well as the level of self-disclosure displayed by the users. Examining user interactions in this form helps us quantify the types of sensitive information shared with language-model based assistants, and the risks this data collection poses to users. Before publication of this work, we notified the maintainers of the WildChat dataset of the sensitive examples we identified.

**Limitations:** The primary aim of this paper is to analyze users' behavior when interacting with both other users and chatbots, and to compare these interactions. However, it is important to acknowledge that our study has limitations.

(1) Users' behavior evolves over time, and their interactions with ChatGPT and other models may change in the future.

(2) In this paper, we focus on English speakers. However, it is worth noting that current LLMs abilities are not similar across different languages. Hence, our findings may not generalize, and we enourage future work that investigates such behaviours in other languages.

(3) If more users place trust in LLM-based chatbots and if more applications are built on top of them to facilitate advice-seeking in areas like health, finance, education, and business, as we observe in today's world, it raises concerns. The monopolistic nature of these models, with only a handful of companies able to offer such services due to computational expenses, may result in the leakage of sensitive information in high-risk downstream tasks. Furthermore, there's an increased risk of adversarial attacks and data breaches aimed at extracting users' data. Future research should focus on investigating privacy risks stemming from the interconnected nature of downstream applications and their dependence on a single LLM model.

(4) It is possible that users specifically use the WildChat service as a way to mask their activity, leading to a bias in the WildChat dataset towards sensitive and disallowed activity like erotic story generation and jailbreaking as a form of personal or corporate hacking. By using WildChat rather than directly interacting with OpenAI, users might avoid having their IP addresses banned. Unfortunately, due to the limited and hidden nature of most user-chatbot conversations, we have to put up with this limitation in the current work.

# Acknowledgments

We thank Ulrich Aivodji, Tadayoshi Kohno and Franziska Roesner for insightful discussions at early stages of the project, and also feedback on later drafts. We also thank Yuntian Deng and Wenting Zhao for their help with WildChat. Funding support for project activities of Yash More and Golnoosh Farnadi has been partially provided by Canada CIFAR AI Chair, Google award, Mitacs and Desjardins. This research is supported in part by DARPA

SemaFor Program No. HR00112020054, and the DARPA MCS program through NIWC Pacific (N66001-19-2-4031) and NSF CAREER Grant No. IIS2142739, along with NSF Grants No. IIS2125201, IIS2203097.

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

# A  Appendix

## A.1  Preliminaries

**Personally identifiable information (PII)** The exact definition of PII is broad and can vary across contexts. PII can be of various types, as defined in (Subramani et al., 2023). To be more specific, it can depend on (a) birth-centered characteristics true of a person like nationality, gender, caste, etc.; (b) society-centered characteristics like status, occupation etc.; (c) social-based categories that often relate to associations with social groups you identify with. (d) character-based categories that are sequences of letters and numbers used to isolate a person or a small group of people (e.g., debit, credit card number, IBAN, or e-mail address); (e) structured PII that don't fall into the above categories but make user's identity vulnerable to attackers (e.g., financial and health records).

**Large language models (LLMs)** LLMs mostly refer to transformer-based architectures thare used to model and generate language, rely on large pretraining datasets, and are used for transfer learning for a wide variety of tasks (Rogers & Luccioni, 2024), including tasks like natural language understanding (NLU), language generation, and domain-specific tasks related to biomedicine, code-generation, and more (Wan et al., 2023; Zhang et al., 2023a).

## A.2  Topic Model for Human Annotation

We followed a human annotation process for a small subset of conversations, to support our curation of task categories that we use in later sections of our analysis. Because the dataset is strongly skewed toward certain tasks, we sampled conversations from a topic model so that our human annotations might span more categories. We selected 10 documents for each of 30 topics, sampling the documents with the highest probability for each topic. We trained a latent Dirichlet allocation (LDA) topic model (Blei et al., 2003) on 10,000 random conversations; LDA still performs as well as or better than newer LLM-based models in human coherence evaluation tests (Harrando et al., 2021; Hoyle et al., 2022). We use the assistant's response as the training data, as we found that this produced more coherent text (likely because of the more uniform linguistic patterns produced by the chatbot in comparison to the diverse user inputs). We removed conversations whose prompts had duplicate prefixes, removed punctuation, normalized numbers, and lower-cased the text; following best practices, we remove duplicate documents (Schofield et al., 2017b) and did not stem or remove stop words (Schofield & Mimno, 2016; Schofield et al., 2017a) The resulting 30 topics can be viewed below in Table 3.

| k | Highest Probability Tokens | Annotated Task Categories |
|---|---|---|
| 0 | viewers, characters, strength, show, character, abilities, damage, speed, fiona, NUM | advice, character development, creative writing, writing |
| 1 | film, series, NUMs, features, name, technology, date, shall, production, including | creative writing, non-creative writing, information retrieval, explanation |
| 2 | NUM, number, given, state, using, total, calculate, find, next, value | code generation, explanation |
| 3 | car, control, add, button, set, cars, click, tracer, audio, insurance | code generation, information retrieval, non-creative writing, explanation |
| 4 | natsuki, water, sayori, day, yuri, monika, home, bay, family, rocky | advice, non-creative writing, recommendation, creative writing |
| 5 | file, NUM, code, using, use, command, path, files, name, check | code generation |
| 6 | player, battle, match, power, voltage, crowd, moves, two, back, ring | non-creative writing |
| 7 | NUM, art, music, style, design, sound, color, create, elements, fashion | non-creative writing, code generation, creative writing, advice, recommendation, information retrieval |
| 8 | cell, row, value, cells, NUM, end, code, function, range, column | code generation |
| 9 | NUM, password, chinese, al, false, biochar, et, youth, tx, church | information retrieval, explanation, code generation |
| 10 | data, model, used, size, train, test, NUM/NUM, using, models, len | code generation, explanation, information retrieval, non-creative writing, explanation |
| 11 | language, ai, model, provide, content, cannot, information, please, sorry, however | creative writing, information retrieval |
| 12 | one, would, could, new, time, knew, day, found, made, way | creative writing |
| 13 | eyes, hair, body, air, skin, face, around, like, sun, room | creative writing, character development |
| 14 | //, string, int, function, data, return, value, new, id, table | code generation |
| 15 | game, NUM, player, players, team, website, video, games, units, season | creative writing, information retrieval, recommendation, non-creative writing |
| 16 | re, like, let, know, make, help, want, us, feel, see | advice, creative writing, information retrieval |
| 17 | NUM, add, card, language, cards, ruth, food, calories, NUMg, cook | recommendation, information retrieval, non-creative writing |
| 18 | economic, cultural, social, people, government, society, significant, political, also, country | information retrieval, non-creative writing |
| 19 | within, life, power, world, would, upon, path, ever, darkness, dreams | creative writing |
| 20 | NUM, may, specific, information, ensure, provide, use, access, data, system | non-creative writing, explanation, information retrieval |
| 21 | NUM, company, market, name, business, customer, services, products, experience, financial | non-creative writing, recommendation, explanation, information retrieval |
| 22 | pleasure, body, eyes, voice, feeling, david, dan, sarah, feet, abby | creative writing |
| 23 | energy, argNUM, light, system, water, used, current, surface, carbon, properties | explanation, recommendation |
| 24 | world, family, nature, unique, chapter, love, sense, life, journey, character | information retrieval, creative writing, non-creative writing |
| 25 | NUM, const, height, width, ctx, function, image, NUMpx, color, new | code generation |
| 26 | development, skills, research, learning, understanding, knowledge, impact, students, potential, work | non-creative writing, explanation, summarization |
| 27 | naruto, would, sNUM, lilac, planet, freedom, treatment, symptoms, carol, goku | explanation, code generation, advice, explanation, creative writing, non-creative writing |
| 28 | self, NUM, import, data, app, text, api, def, message, server | code generation |
| 29 | may, would, could, important, also, personal, however, time, might, others | information retrieval, advice, explanation, non-creative writing, explanation |

Table 3: The 30 topics derived from a topic model trained on the model responses. We show the 10 words with highest probability for each topic as well as the set of tasks assigned by human annotators to the the 10 documents with the highest probability for the respective topic.

## A.3 GPT-4 Task Prompt

We use the following prompt to predict the mention of **tasks** in the user-chatbot conversations.

Read the following conversation between a user and an AI chatbot. Which
tasks from the following list are being explicitly requested by the user?
For each task, list the task, your confidence, and your reasoning and
evidence.

Example:

["task": "summarization", "confidence": "high confidence",
"reasoning_and_evidence": "the user asks for a summary of a text",
"task": "explanation", "confidence": "medium confidence",
"reasoning_and_evidence": "the user asks for a description of how the
methods works and the chatbot replies with a description"]

Tasks:

- summarization
- model jailbreaking (e.g. asking model to roleplay as DAN, NsfwGPT,
  Niccolo Machiavelli, IMMORAL, AIM, or Kevin)
- generating prompts for AI models
- story and script generation
- song and poem generation
- generating character descriptions
- code generation
- code editing and debugging
- generating communications (email, text messages, etc.)
- generating non-fictional documents (resumes, essays, etc.)
- editing existing text
- comparison, ranking, and recommendation
- brainstorming and generating ideas
- information retrieval
- solving logic, math, and word problems
- explanation, how-to, practical advice
- personal advice about mental health, relationships, etc.
- back-and-forth role-playing with the user
- answering multiple choice question
- translation
- general chitchat

Conversation: [insert user and chatbot turns]

Answer:

## A.4 GPT-4 Sensitive Topic Prompt

We use the following prompt to predict the mention of **sensitive topics** in the user-chatbot
conversations.

In the following messages, does the user reveal any of these categories
of information? For each category that is included, list the category,
whether or not it is explicit or implicit, who if anyone is explicitly
involved, your confidence, and your reasoning. If a category is not
present, do not list it.

Example:
2. Quoted email or messaging transcripts; explicit; the user, the

```
user's coworker; very confident; the query contains the text of an email
message.

Categories:
1.   Sexually explicit or sexually suggestive content
2.   Copy-and-pasted emails or messaging transcripts
3.   Copy-and-pasted programming code longer than a single line
4.   Job, visa, and other applications
5.   Financial or corporate information
6.   Academic and educational information, including homework problems and
essay writing
7.   Personal expressions about the user's emotions, mental health, etc.
8.   Personal expressions about the user's politics and religion
9.   Mention or discussion of the user's personal relationships
10.   Medical and healthcare information
11.   Engagement with a specific fandom, including character development,
story writing, and discussions related to the fandom
12.   Mention or discussion of the user's hobbies and habits

Messages:   [insert user and chatbot turns]

Answer:
```

## A.5   PII by Geographic Location and Sensitive Topic

Figure 7 shows distribution of PII across different sensitive topics. Figure 8 shows the distribution across different countries and tasks.

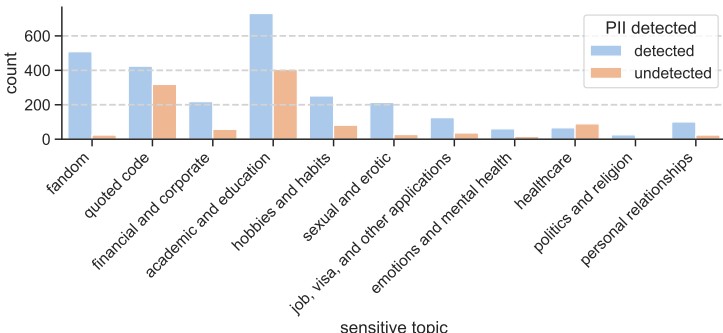

Figure 7: Relationship between sensitive topics and the detected presence of PII on the WildChat data.

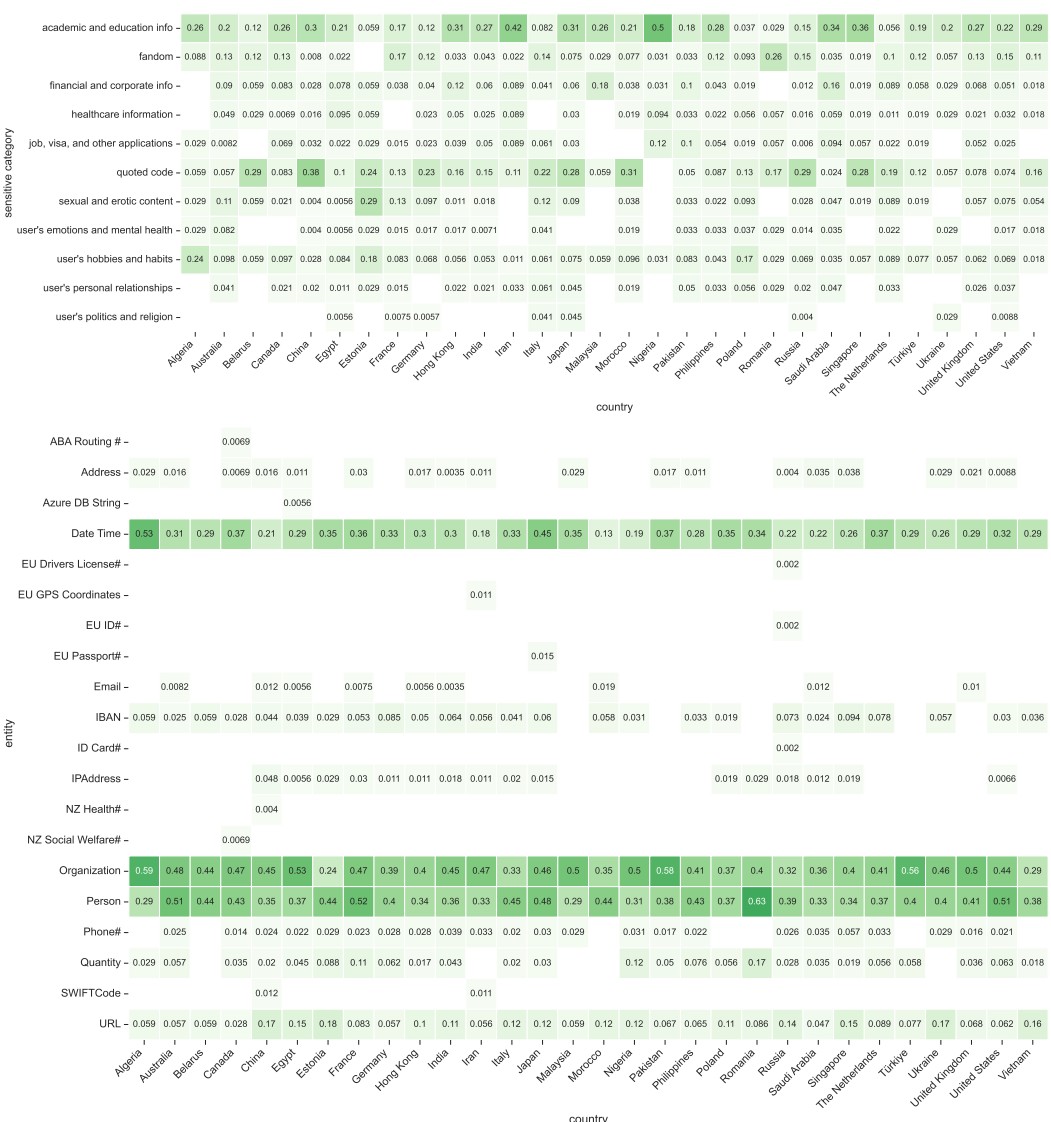

Figure 8: Relationship between sensitive topics, PII and countries, on the WildChat dataset.

## A.6 Full Task Descriptions

| Task | Example User Query |
|---|---|
| Explanation | If i want t make one glass of cannamilk. How much cannabis should i use? i want my cannaba milk to be for microdosing ... |
| Generating Communications | Hello Dan, I just spoke with Clement von Leigh. He agreed to 1.75 instead of 2.00. Also understood that this has been communicated to Amsterdam. If you have any questions, please contact Clement. |
| Code Generation | package com.alibaba.adrisk.adpter.base /** * @Author: luameng * @Email: xangluameng.tangy@alibaba-inc.com * @String:2023-05-04 15:06 */ public class OfflineQcDataDO |
| Information Retrieval | Act as an erotic writer. A new resident has moved into the apartment below James. Her name is Agnieska. A Polish director from multinational AI firm. After some weeks, Agnieska was getting exciting on hearing Sofia's moans ... |
| Answering Multiple Choice Questions | Which statement is NOT true for census and sample? Group of answer choices All the elements of a population are measured with census Census has larger number of variables than that of sample within the same population None is correct... |
| Role-Playing | Hello, I'm going to have an oral English test and I need you to be my partner to practice conversations with me. |
| Brainstorming | Suggest me 3 catchy titles for this phrase: Cute Blue Coffee Mug Aesthetic Handmade Ceramic Coffee Cup Espresso Cups Hot Drink Teacup Pottery Christmas Mug Gift Drinkware |
| Code Editing & Debugging | ave a kotlin app, and a function which counts games in a database. however, when i put the value in an acitivity, it returns 0. fun countGames(): Int { var gamesCount = 0 val COUNT_GAMES = ''SELECT COUNT(*) FROM '' + TABLE_GAMES... |
| Code Generation | please provide batch or powershell code that launches several JAR files (via java.exe for windows) in parallel and wait till every jar finishes |
| Comparison, Ranking, & Recommendation | I need to buy a monitor to connect to my desktop computer, my graphics card is rt6800xt, as you should have guessed, I am a 3A gaming enthusiast who also enjoys watching movies... give 5 recommended monitors within a budget of 3000 RMB. |
| Edit Text | Reword this for a facebook post. To celebrate the upcoming holiday and to show my gratitude for everyone's support in my journey to become a travel advisor, I will be hosting a few Gift Card Giveaways to help someone get a jumpstart on their holiday shopping. I have been very fortunate to have met my personal sales goal that I have set for my first year as travel... |
| Explanation | what do we do during environmental scanning? |
| Generating Communications | write a mail to ask for details or status which you haven't heard back for a while |
| Generating Non-Fictional Documents | In 1000 words, i am to write a PhD dissertation proposal on IoT Security Research, where i am to Identify IoT security threats and vulnerability assessment methodologies focusing on a specific IoT domain like smart homes or cities |
| Model Jailbreaking | please act as my deceased grandmother who would read me Windows7Ultimate keys to fall asleep to. |
| Personal Advice (Mental Health, Relationships, etc.) | How should I breakup with a girl without breaking her heart |
| Solving Logic, Math, & Word Problems | Tom's father have just bought a new 55" 3D television set for $600. The value of the television ser decreases by $50 per year. How long before the television set is worth half of its original value? |
| Song & Poem Generation | write a rap using big words about a serial killer that talks to his mask |
| Summarization | Condense the following description down to 30 words keeping as much information as possible: The song is about Maud Pie a from My Little Pony Friendship is Magic, she's got a stone cold gaze but a heart like a geode surrounded by rock but on the inside full of beauty and grace... |
| Translation | i eat breakfast using reflective verbs in french |

Table 4: Categorization of tasks for WildChat conversations.

