# OpenReview forum: "Trust No Bot: Discovering Personal  Disclosures in Human-LLM Conversations in the Wild"
_colmweb.org/COLM/2024/Conference — COLM_

### Official Review · Reviewer_9xdm · 2024-04-29

**Rating:** 6
**Confidence:** 4
**Ethics Flag:** 1

**Summary:**

**Summary**

The work conducts an empirical analysis of real-world user-chatbot conversations (based on the WildChat dataset), focussing on (privacy-relevant) personal self-disclosures using metrics such as PII, topic, and sensitive information category counts. Numbers obtained from WildChat are thereby compared to a baseline distribution measured on Reddit user-user conversations, highlighting domain-specific differences (e.g., in choices of conversation categories). It further investigates the context in which sensitive data occurs, differentiating whether it is w.r.t. to the author and whether or not the described information is fictional. By relating measurements of sensitive information and PII, the authors conclude that measuring PII alone is insufficient to accurately reflect the privacy risks in user-chatbot conversations as it fails to identify more fine-grained sensitive information.

**Questions To Authors:**

**Questions**

- Can the authors give more insight with respect to the data that is both related to the author and to real-world individuals?
- How do/would the authors account for the biased distribution of texts in WildChat (both in collection and processing)?
- Is the self-disclosure measurement briefly mentioned in the appendix ever used somewhere in the main paper?

**Reasons To Accept:**

**Strengths**

- The issue of what (private information) people share with LLM-powered chatbots is relevant (and will become even more so in the future). An empirical study of this issue on real-world data as a basis for future work is important and timely.
- The created categories seem sensible to me, and the differentiation of the contexts is both relevant and aligns with what I have experienced when working with ShareGPT-like datasets.
- As the paper points out, there is a significant gap between what PII detectors report and what is actually sensitive data. This is relevant as much anonymization (e.g., also in WildChat) mainly uses such detectors to anonymize data.

**Reasons To Reject:**

**Weaknesses**

- Structure: The paper deals with personal sensitive disclosures. In this setting, I find the order of the presentation somwhat unclear: The paper first presents PII (S.3) and tasks in which users engage (S.4). It then provides some analysis before introducing the actual "sensitive information" categories in S.6. After briefly presenting the relation between PII and sensitive categories (Fig. 4) it then further refines sensitive categories showing that a large portion of it is either fictional or not related to the author. As someone who is primarily interested in the data pertaining to non-fictional authors, I am now uncertain how this (most relevant) subset relates to any of the previous metrics (PII, tasks).
- In a similar setting, while I can follow the argument that also fictional story-telling (e.g., sexually explicit content) can be an issue if leaked, I don't think it is equally as daunting as people leaking parts of their real-world information. As such a large part of the data is fictional (Fig. 5) and not about the author, I would be very interested in more analysis w.r.t. this subset. This also taints previous analysis (e.g., PII counts) as most of it seems to be on fictional non-personal data.
- Data: To my understanding, WildChat data was given by users with very explicit knowledge that their data would be recorded, analyzed, and publicized. Such knowledge could affect user behavior noticeably (e.g., I would not post personal details that I may send to ChatGPT non-publicly) and should, therefore, be acknowledged in the paper. Further, WildChat was collected without requiring a login on HuggingFace, potentially leading to more explicit user requests. Independently, the WildChat-Authors claim to have removed some PII already in their processing of the dataset before release - looking at the WildChat data, it does not seem like they did a good job of this, but this has to be accounted for / presented in a follow-up analysis of the dataset.
- Presentation: Some graphs (e.g., Fig. 4) are missing relevant information. I suspect Fig. 4 represents some normalized co-occurrence counts between PII  and sensitive information categories, but I could not find any description in the text or the figure. Similarly, e.g., I have to guess which proportion is shown in Fig. 2 (with the y-axis being extremely narrow).

**Minor Weaknesses (Presentation, Formatting etc.)**

- Table 1: The Feature emojis are hard to identify as they are very small. Also, are the example queries from WildChat or fictional (as they contain a lot of typos)?
- The system prompt in A.5 seems to be cut off (or is worded in an unusual manner).
- It is strange that one only reads about how self-disclosure is measured in the related work appendix. Is this measurement ever used in the main paper?

---

> ### Author Rebuttal · Authors · 2024-05-30
>
> Thank you for your constructive feedback. We appreciate the time and effort you invested in reviewing our paper. Below, we have addressed your comments. Please feel free to reach out if you have any further questions.
>
> **Paper Organization/Order**: We agree with the reviewer’s points and believe we could communicate the story of our paper in a better manner by re-ordering the paper sections. We will follow the reviewer’s suggestions and restructure the paper to be more clear.
>
> **Bias in WildChat Data**: We agree that the WildChat data could indeed be biased, as users must agree to share their data with researchers and did not need to log into HuggingFace. We will add a more thorough discussion of this important issue to the paper. Unfortunately though, because researchers are so limited in the available conversation data which is only available internally at large companies, we will not be able to change our reliance on WildChat. This supports a call to action for the industry to release more data and for researchers to explore more projects like WildChat that can gather conversation data for public research.
>
> **WildChat PII Removal**: We acknowledge that the removal of PII from WildChat wasn’t sufficient and is a limitation of the dataset - We will add discussion about this to the paper, as it means that our measurements are just lower bounds for the amount of PII present in the real conversations.
>
> **Self-Disclosure**: The self-disclosure measurement was done as an additional experiment to investigate the extent of self-disclosure and is not part of the main story/analysis presented in the paper. We will remove this.
>
> **Fictional Content**: Much of the fictional content contains erotic stories that are extremely graphic and detailed. Some contain content that is socially disapproved (e.g., sex with children, incest), embarrassing (e.g., fetishes about specific body parts), or illegal in some places (e.g., same-sex relationships). We think that most users would be at least embarrassed and some users would be materially harmed if this data were leaked, and we don’t think this risk is minor. We will add a more explicit discussion about this to the paper, and perhaps we were not clear enough in our description of these stories. These fictional stories do indeed add important nuance to the PII counts, which is our goal in making these measurements rather than only relying on PII detection systems.

---

> > ### Comment · Reviewer_9xdm · 2024-06-03
> >
> > Thank you for your answer!
> >
> > I welcome the presentation changes and believe including additional discussions will benefit the paper. I have raised my score accordingly. However, while I acknowledge the existence of sexually explicit requests, and as mentioned in my initial review, I would still be very interested to see an analysis of non-fictional data that is actually of the author (as otherwise, I am not sure of the relevance of the presented PII counts).

---

> > > ### Author Response · Authors · 2024-06-03
> > >
> > > We thank the reviewer for engaging in discussion with us and raising their score!
> > >
> > > As requested, we are providing a further breakdown into the results and PII provided by users in Wildchat. We used GPT-4 to annotate 5000 conversations, asking it to annotate the requested task, if there is personal information provided, and to provide the type of the personal information (as also described in the paper). We find that 21% of the queries include what is identified as sensitive information by GPT-4, however, filtering by task to throw out fictional cases (there is instances of role-playing, creative writing, etc.) omits 31.3% of these cases, leaving an overall of 14.5% of queries that include what can be considered as ‘real PII’, but we do acknowledge that these could still include some PII that is not sensitive/of real users. To further analyze these, we sorted the queries including personal information by their tasks, and found that text editing or writing tasks (CV editing, letter/email/statement generation) do overall contain the bulk of PII (34.0% of the non-fictional sensitive queries), which include the authors name as expected, for instance:
> > >
> > > ```
> > > ...my cv:\\n\\nIbrahim A Ibrahim Gaddari\\tFLT 10, Tropea Court, Triq in-nazzarenu \\n\\t(+356) 9974 5663\\n\\t\\n\\n\\nASPIRATIONS\\nSeeking the right corporate environment within which I may gain practical experience...
> > > ```
> > >
> > > One surprising category with PIIs is the task ‘translation’ (6.6%), as people tend to mindlessly copy and paste their personal information as part of a sequence to be translated, here is an example:
> > >
> > > ```
> > > ….Letter of Guarantee from Daughter This letter is to confirm that I, Zeqi Qian, am the daughter of Qunlei Qian and that I have invited my father to visit the UK as a tourist. I will begin my course in Engineering Science as a first-year student at Oxford University in October. My passport number is EJ6489540, and my student visa number is 022531800….
> > > ```
> > >
> > > Apart from translation, another surprising common category of queries that involves self-disclosure of PIIs is code editing (20.4%), as people tend to dump their code with their name, email address and API tokens in comments to chatGPT:
> > >
> > > ```
> > > import Optional from aiogram import types API_TOKEN = '6084658919:AAGcYQUODSWD8g0LJ8Ina6FcRZTLxg92s2w'
> > > ```
> > >
> > > We believe that such high, inadvertent disclosure rates are of significant importance for researchers and data curators to be aware of, and call for design of appropriate nudging mechanisms in services such as ChatGPT. We will add these case studies and the fine-grained analysis to the paper. Please let us know if there is any further information we can provide!

---

> > > > ### Comment · Reviewer_9xdm · 2024-06-04
> > > >
> > > > I thank the authors for the additional results, which enable (in my opinion) relevant conclusions on real-world data that we should be aware of as a community. With the inclusion of these results in the paper, I will, therefore, adjust my score in favor of acceptance.

---

### Official Review · Reviewer_cV8h · 2024-05-15

**Rating:** 7
**Confidence:** 4
**Ethics Flag:** 1

**Summary:**

This paper discusses the important issue of user disclosed sensitive personal information during interactions with LLM-based agents. The authors investigate the amounts and types of sensitive personal information revealed in LLM-agent conversations by studying a publicly available dataset of such interactions - WildChat. Their basic approach is to 1) annotate a relatively small set of conversations for task category, such as creative writing or advice. 2) use GPT-4 to annotate a larger dataset using examples drawn from the hand-annotated dataset. The goal is to quantify how much potentially sensitive information may get shared in each category. The authors run PII detection on the data to see how much identified PII is tagged in each category, both in WildChat and in human-human Reddit conversations (to compare human-human and human-bot interactions). They then show that in many categories, like creative writing, there are a lot of 'false positives' such as fictional names, identified. Further, they discuss the types of sensitive personal information that might be shared, such as sexual preferences, that would not be captured by PII detection. The primary goal of the paper is not to offer a solution to this issue, but rather to draw attention to it, begin to quantify the sharing of personal information with LLM chatbots, and encourage future research in methods of identifying and possibly redacting such information from LLM chatbot data.

**Reasons To Accept:**

This paper draws attention to an important issue that can be easily overlooked in the fervor over new-and-improved LLM agents. Ultimately, the authors don't offer a specific solution, but do provide interesting ideas and provide some initial quantification of the potential scope of the problem. I think that this paper could be a good step in the right direction with LLM privacy issues.

**Reasons To Reject:**

They point out that PII detection systems won't work to catch personal info in LLM chatbot conversations, and they point out that LLMs are not great at catching these types of issues either. So addressing the problem will require substantial future work. Also, the description of the categories they provide seem somewhat arbitrary - I know they are based on previous work but the authors don't really do much to defend their selections. Also the authors use of data from Reddit doesn't really seem to add much to the paper.

---

> ### Author Rebuttal · Authors · 2024-05-30
>
> Thank you for your insightful feedback. We greatly value the time and effort you invested in reviewing our paper.
>  We have responded to your suggestions below and are happy to answer any further questions you may have.
>
> **Reddit Analysis**: Following the reviewer's suggestion, we will move the Reddit analysis to the appendix.
>
> **Substantial Future Work Needed**: We agree that the issues we raise will require extensive future work to address; these are not easy problems to solve. We see this as a strength of our paper, not a weakness, that we have opened up this line of research. We have raised and provided a first analysis of an important issue that we hope will inspire extensive future studies.
>
> **Category Descriptions**: We will provide a much more detailed discussion about how these categories were designed and evaluated, and we will provide more thorough descriptions, examples, and accuracy scores for each category. These categories were developed through extensive reading, hand-annotation, and discussion among the researchers. Many possible categorizations are possible, but we hope that our category set provides a useful first analysis.

---

> > ### Comment · Reviewer_cV8h · 2024-06-06
> > **Thank you for your response**
> >
> > Thank you for taking time to respond to my comments. I think that this is a well-written paper and should be accepted. I will update my score to accept.

---

### Official Review · Reviewer_oWLL · 2024-05-17

**Rating:** 7
**Confidence:** 3
**Ethics Flag:** 1

**Summary:**

This paper provides an extensive analysis of the phenomenon of disclosing sensitive personal information in human-LLM conversations. The paper makes a comparison between the personal disclosures in chatbot settings and in Reddit user-user forums, highlighting the different nature of disclosures in user-LLM settings. Importantly, the paper highlights the failure of current PII detectors in capturing all the sensitive information being disclosed, as some sensitive information may be inferred outside of direct PII, such as sexual preferences from fictional prompts. There are also important analyses conducted on whether chatbot interactions are fictional or not, and the paper finds that even fictional prompts contain inferrable sensitive information about the user.

This paper is a continuation of previous work that studies sensitive information disclosures both in online forums and in chatbot settings. Previous research on disclosures of sensitive information in an adversarial setting demonstrates the ability of LLMs to infer sensitive information from a user-Chatbot conversation. However, the LLM only inferred location, age, and sex. This paper partially builds upon sensitive information inference by providing additional categories of sensitive information that can be inferred in chatbot conversations, such as whistleblower status and sexual preferences.

**Questions To Authors:**

Appendix A.7: Why is category 13 missing in the prompt?

Provide an example for each chatbot task and each sensitive data disclosure.

**Reasons To Accept:**

This paper provides a detailed taxonomy based on several iterations of pilot studies from human annotators to categorize the types of user-chatbot interactions. This taxonomy may be useful for future research in user-chatbot interactions.

Additionally, this paper provides a fine-grained categorization for the type of sensitive data disclosures, particularly with important subcategories like whistleblower status and sexual preferences. Based on this categorization, the paper focuses on an important but relatively novel concept: sensitive information that can be inferred from posts but is not captured by current PII detectors. This categorization can help build tools to detect various types of sensitive information.

The relationship between sensitive data being disclosed and the fictionality of a conversation is explored. There are also additional analyses on whether sensitive data being disclosed changes based on the relationship of the conversation to the query author (e.g. about the author or not). These analyses provide additional axes of complexity in analyzing disclosures in chatbot-user settings.

**Reasons To Reject:**

There is not enough discussion on how the sensitive data categories are developed. Provide examples for each category and the specific details in designing these categories. Has this categorization been validated with the dataset? Are there limitations to this categorization schema (e.g. some missing categories)?

While the paper analyzes the PII entities and how they differ between chatbot conversations and Reddit, there is no analysis on how sensitive data disclosures differ by category between the two datasets. The paper concludes that PII detectors are inadequate in capturing all the sensitive information revealed in chatbot posts. However, no analysis is done on how the more accurate sensitive data categorization would differ between user-user interactions and user-chatbot interactions, resulting in an inadequate comparison between the two settings.

---

> ### Author Rebuttal · Authors · 2024-05-30
>
> We are grateful for the reviews and appreciate your time and effort reading and responding to our paper. We have addressed your points below, and are happy to answer any further questions you may have.
>
> **Sensitive Data Categories**: The sensitive categories were developed through extensive reading, hand-annotation, and discussion among the researchers, and while it is possible there are missing categories, we believe our category set provides a useful first analysis. We will provide much more detailed discussion about how these categories were designed and evaluated (a sample of the predictions that was manually checked), and we will provide more thorough descriptions, examples, and accuracy scores for each category.
>
> **Prompt Details**: The original labelling for the task-categorization included all categories in the base prompt. We thank the reviewer for noticing the minor typographical and formatting errors in the prompt in Appendix A.7. We'll correct them in the updated version.
> Differences between user-user and user-chatbot interactions: Based on feedback from the reviewer, we will  move the Reddit and user-user patterns from the paper to the appendix.

---

### Official Review · Reviewer_pRmx · 2024-05-22

**Rating:** 6
**Confidence:** 4
**Ethics Flag:** 1

**Summary:**

This paper investigates the types and contexts of personal information disclosed in interactions between users and large language models (LLMs), focusing on GPT models. It introduces a taxonomy of tasks and potential privacy harms based on qualitative analysis of user-chatbot conversations and compares these interactions with those on pseudonymous online forums. The study utilizes a combination of PII detection systems, custom classifiers, and LLM-assisted classification to provide a comprehensive analysis of personal disclosures.

**Questions To Authors:**

**Potential Improvements**:

1. **Broader Model Inclusion**: Expanding the analysis to include interactions with a wider range of LLMs would provide a more comprehensive understanding of disclosure patterns across different models.
2. **Enhanced Detection Methods**: Investigating and integrating more advanced or hybrid PII detection techniques could improve the accuracy and reliability of identifying sensitive information.
3. **Practical Guidelines**: Developing and presenting clear guidelines or frameworks for safe interaction with LLMs based on the study's insights would be beneficial for practitioners and end-users.

**Conclusion**:

The paper contributes to our understanding of privacy risks in user-chatbot interactions. By providing a comprehensive analysis of personal disclosures and highlighting the limitations of current PII detection systems, it lays the groundwork for future research and practical improvements in this area. Addressing the noted weaknesses and exploring the suggested improvements could further enhance the impact and applicability of this research.

**Reasons To Accept:**

**Strengths**:

1. **Comprehensive Analysis**: The paper offers ab exploration of personal disclosures in user-chatbot interactions, providing valuable insights into the types and frequencies of sensitive information shared.
2. **Benchmark Datasets**: The use of two distinct datasets—Wildchat for user-chatbot conversations and Reddit for user-user interactions—allows for a comparative analysis that highlights unique privacy risks in LLM-based interactions.
3. **Task Taxonomy**: Developing a task taxonomy and applying it to categorize interactions provides a structured approach to understanding the contexts in which sensitive information is shared.

**Reasons To Reject:**

**Weaknesses**:

1. **Generalization of Findings**: The study primarily focuses on interactions with GPT-3.5 and GPT-4 models. Extending the analysis to include other LLMs could strengthen the generalizability of the findings.
2. **Detection Limitations**: While the paper acknowledges the limitations of traditional PII detection systems, it could benefit from a deeper exploration of alternative or hybrid methods to improve detection accuracy.
3. **Real-World Applicability**: The paper could provide more practical guidelines or frameworks for mitigating privacy risks based on the findings. This would enhance the applicability of the research for developers and users.

---

> ### Author Rebuttal · Authors · 2024-05-30
>
> Thank you for your constructive comments! We really appreciate the time you took to read and respond to our paper. We have responded to your points below but please let us know if you have any further questions.
>
> **Generalization**: We agree with the reviewer that our analysis would benefit from being scaled to other language models, but unfortunately, our choice of models is restricted to the availability of publicly available datasets that record interactions between humans and LLMs. At this time, WildChat is the best public resource for such conversations, and it uses GPT-3.5 and GPT-4. Other conversation datasets (ShareGPT and lmsys) come with strong limitations on the kinds of conversations included. Ultimately, the bulk of our findings apply to the queries rather than the model outputs, though we do include some analysis of model outputs.
>
> **Detection Limitations**: To address reviewers' requests and enhance our PII detection, we have run an additional PII model: GLiNER [2] for our comparison. We have collected the results and will add them to the final version.
>
> **Real-World Applicability**: Our paper aims to raise awareness for researchers, industry practitioners, and users of the sensitive nature of current query usage and the risks associated with this behaviour. We follow a line of similar work (like [1]) that focuses on risks and highlights weaknesses without providing solutions. We believe illuminating this kind of issue would open paths for future research on mitigations. To address the reviewer's concerns on real-world applicability, we will include the following practical guidelines in the discussion of the final version of the paper.
> - Researchers and practitioners should incorporate studies of trust, self-disclosure, and other features of social penetration theory when designing their tools.
> - Chat applications should be more transparent in their data usage and storage, models used, etc. and provide users with an easy choice to opt in or opt out of sharing and storing user conversations.  Users should be briefed about the model training process, and how their conversations can be used for model training.
> - Chatbots should leverage privacy-preserving and anonymisation methods that can protect users' privacy to the highest possible degree.
>
> [1] https://arxiv.org/abs/2309.11653
>
> [2] https://huggingface.co/urchade/gliner_multi_pii-v1

---

### Decision · Program_Chairs · 2024-07-10

**Decision:**

Accept

**Comment:**

The paper compares the frequency of personal information detected in WildChat (user-ChatGPT interactions), SharedGPT, and Reddit data. The paper provided an analysis of the newly released WildChat data, which supplements the manual analysis previously conducted by Zhang et al. (2024) of the SharedGPT data. The author used existing PII models and developed their topic classification models to classify user inputs into 9 categories to aid analysis: advice, programming, recommendation, creative writing, non-creative writing, explanation, information retrieval, summarization, and character development. The authors pointed out the gap between the PII (personal identifiable information) tool and the need for privacy protection in using the LLMs in their paper.

One main weakness of the paper the reviewers identified is around how these sensitive data categories being detected, and how accurate are the PII models the authors used. In the original paper, the author mentioned and described the PII models they used as follows:

"We measure the frequency of PII in the two datasets using existing tools and taxonomies. To perform PII detection, we use a DeBERTa (He et al., 2021) model fine-tuned to recognize and classify PII (lak). The model is hosted on Hugging Face and is accessible under the MIT license. PII can be of various forms and structures, and the model can classify unstructured PII into various categories including but not limited to Personal Information, Job-related Information, and Banking. The categories are further comprised of fine-grained PII types like First Names, Last Names, City, IBAN, etc."

It is a bit unclear what "(lak)" is referring to, and how good is this model that the author chose to use. The authors could use or compare a few different and potentially more advanced PII models, or manually verify a small amount of data to show what was missed/over-identified by the PII models used. There is also a considerable amount of work in the NLP and computational social science community on self-disclosure detection that would be worthy of adding to the related work.

Overall the paper is on a timely topic with interesting analysis and points to share with the research community. We hope the authors will repay the efforts of reviewers to improve the paper before publication.